# Application of a Novel Picard-Type Time-Integration Technique to the Linear and Non-Linear Dynamics of Mechanical Structures: An Exemplary Study

Evgenii Oborin 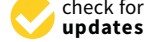 and Hans Irschik *

Institute for Technical Mechanics, Johannes Kepler University Linz, Altenbergerstraße 69, 4040 Linz, Austria;
evgenii.oborin@jku.at
* Correspondence: hans.irschik@jku.at; Tel.: +43-732-2468-6307

**Abstract:** Applications of a novel time-integration technique to the non-linear and linear dynamics of mechanical structures are presented, using an extended Picard-type iteration. Explicit discrete-mechanics approximations are taken as starting guess for the iteration. Iteration and necessary symbolic operations need to be performed only before time-stepping procedure starts. In a previous investigation, we demonstrated computational advantages for free vibrations of a hanging pendulum. In the present paper, we first study forced non-linear vibrations of a tower-like mechanical structure, modeled by a standing pendulum with a non-linear restoring moment, due to harmonic excitation in primary parametric vertical resonance, and due to excitation recordings from a real earthquake. Our technique is realized in the symbolic computer languages Mathematica and Maple, and outcomes are successfully compared against the numerical time-integration tool NDSolve of Mathematica. For out method, substantially smaller computation times, smaller also than the real observation time, are found on a standard computer. We finally present the application to free vibrations of a hanging double pendulum. Excellent accuracy with respect to the exact solution is found for comparatively large observation periods.

**Keywords:** mechanical structures; linear and non-linear dynamics; time integration; Picard-type iteration; symbolic computation; tower-like structure; earthquake excitation; single and double pendulum

## 1. Introduction

The need for novel computationally efficient time-integration schemes is increasingly invoked by online and real-time applications of non-linear dynamics of mechanical structures. Developments are rapidly proceeding to extended applications, e.g., in earthquake excited structures, concerning control of structural vibrations, see, e.g., [1,2], hybrid testing, see, e.g., [3], or non-linear real-time hybrid simulation, see, e.g., [4], for a force correction method. For the development of dynaimc structural models in linear non-linear earthquake engineering, as well as for suitable time-integration methods and non-linear control formulations, the reader is moreover referred exemplarily to [5–10]. In the following, a novel efficient time-integration scheme recently derived by the present authors, see [11], is applied for simulating non-linear forced and free vibrations of mechanical structures, and its advantages against standard numerical time-integration methods, such as NDSolve, available in Wolfram Mathematica [12], are exemplarily demonstrated. Our technique consists in an application of an extended Picard iteration to the time-integrated (global balance) form of the equations of motion, see [13] for the original Picard iteration and [14] for the structural equations of motion in integral form, i.e., for the global relations of balance of momentum. Concerning the applicability of this technique, we believe that it will be suitable as long as the derivation of the equations of motion has lead to a system of linear or non-linear ordinary differential equations of second order for the required degrees-of-freedom (DOFs), accompanied by the necessary number of initial conditions.

In structural mechanics, such an initial-value system can be obtained by starting from the relations of (linear and/or angular) momentum for suitably modelled substructures (rigid bodies, deformable finite elements after discretization), taking into account constitutive relations, and using the usual reduction techniques to obtain a system with minimum number of DOFs. The system afterwards must be formally time-integrated to a system of first order. The time-integrated system then represents a system of so-called the global relations of balance of momentum, or first integrals. Often, when possible, it is more advisable to use the latter from the scratch, due to their wider applicability, since only the existence of integrals are to be required from a mathematical point of view, and no differentiability, or bounded integrands. Only mild mathematical assumptions hence must be made for allowing to substitute approximations into the integrands of these global balance forms of the equations of motion to obtain an improved solution, as this was originally suggested by Picard [13]. In the following, in extension of Picard [13], explicit discrete mechanics-type approximations, see Greenspan [15], or Runge–Kutta approximation methods, see [16], are used as starting guess in the Picard iteration, together with some advances that are nowadays offered by symbolic computation tools, e.g., in [12]. A main feature is that, thanks to the power of symbolic methods, our technique can be formulated so that the Picard iteration and the necessary symbolic operations need to be performed only once, before the time-stepping procedure starts. In our previous investigation [11], computational advantages have been demonstrated for large free vibrations of a hanging rigid pendulum, for which exact solutions do exist, and which is used in the literature as a benchmark example for comparison with numerical time integration methods. It was shown in [11] that a large number of free non-linear vibration cycles can be accurately computed, where the simulation remains in the close vicinity of the exact free phase-plane trajectory of the pendulum, due to the algorithmic satisfaction of the relations of global balance. Excellent accuracy with respect to the exact solution and competing numerical schemes was observed, also for large observation periods. In the present paper, we first study the application to the non-linear forced motions of a tower-like mechanical structure in the form of a standing (inverted) rigid pendulum with a non-linear restoring moment under a ground excitation of the earthquake-type and gravity. Structural parameters are taken from a real tower discussed in [17]. For small vibrations of the earthquake-excited pendulum with an inelastic spring considering the P-Delta effect, see [18]. For some other advanced applications of pendulum vibrations in the present context, see, e.g., [19,20]. In the following, we consider two cases of excitation: a harmonic excitation, which in vertical direction is in primary resonance, and a real earthquake excitation from recorded acceleration data. At a smaller scale, the model of a standing pendulum is also of interest for damping ground-excited vibrations of mechanical structures, see [21]. In the present contribution, we utilize results of the fourth-order Runge–Kutta method [16] as guess in the first step in the extended Picard iteration. Very similar time recordings of the non-linear structural response of our method are observed for our method on a standard computer, when compared to the standard initial value problem (IVP) solver NDSolve [12], but with substantially smaller computation times. Since computation times are also much smaller than the real observation periods of earthquake with a long duration, the technique should be of interest for real-time applications, such as in automatic control, particularly also because of the underlying symbolic formulation. In the last part of the paper, as a first step towards applications to non-linear multi-body dynamic systems, we present the application to free vibrations of a hanging double pendulum, for which also exact solutions do exist. In this example, two explicit discrete-mechanics type solutions originally suggested by Greenspan [15] are used and compared as starting guesses in the first Picard iteration. Symbolic computations are performed in Maple [22]. In order to show that the method can also be used conveniently for linear dynamic problems, we restrict to free vibrations, for which closed-form solutions do exist, and concentrate on accuracy aspects here. Excellent accuracy with respect to the exact solution is demonstrated, also for

a comparatively large observation period, which proves that the method can be applied with high advantage for linear vibrations also.

## 2. Rigid-Body Model of an Earthquake Excited Tower-Like Structure

A fundamental non-linear model of a tower-like structure is shown in Figure 1. We regard the tower-like structure as an inverted pendulum with a non-linear spring at its hinge support, which is excited in both vertical and horizontal directions. We denote the total mass of the structure by $m$ and its radius of inertia by $i$. The base accelerations are $a_H$ and $a_V$. The mass center is located at a distance $L$ from the support. We take a non-linear cubic restoring moment for the spring: $M = -k_L \varphi + k_N \varphi^3$. The horizontal and vertical support reactions are called $F_H$ and $F_V$. The unknown degree-of-freedom (DOF) in this 1DOF model is the angle $\varphi$ counted from the vertical direction.

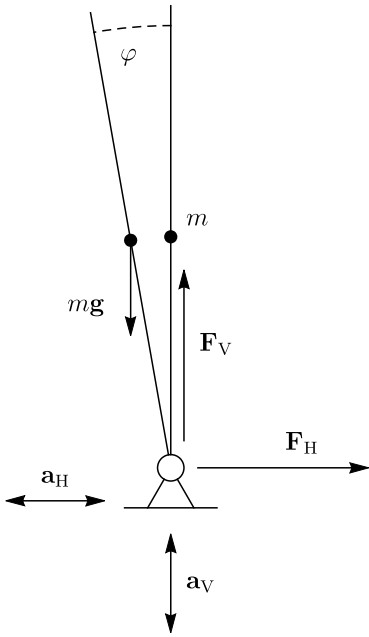

**Figure 1.** Earthquake excited tower-like structure: rigid-body model.

The relation of balance of moment of momentum about the center of mass reads

$$mi^2\ddot{\varphi} = F_V L \sin\varphi + F_H L \cos\varphi - k_L \varphi + k_N \varphi^3. \tag{1}$$

Derivatives with respect to time $t$ are indicated by superimposed dots.
Balance of linear momentum may be written as

$$m\left(a_H - L\ddot{\varphi}\cos\varphi + L\dot{\varphi}^2 \sin\varphi\right) = F_H,$$
$$m\left((a_V + g) - L\ddot{\varphi}\sin\varphi - L\dot{\varphi}^2 \cos\varphi\right) = F_V. \tag{2}$$

We may use the latter relation in order to eliminate the support reactions from balance of moments of momentum Equation (1) and end up with the following non-linear second-order ordinary differential Equation (ODE):

$$m\left(i^2 + L^2\right)\ddot{\varphi} = m((a_V + g)L\sin\varphi + a_H L\cos\varphi) - k_L \varphi + k_N \varphi^3. \tag{3}$$

This relation can be regarded as balance of moment of momentum about the moving support, see ([23], Chapter 7). With the frequency $\Omega_0$ of the freely hanging linear pendulum when $\pi - \varphi \ll 1$, we obtain

$$\ddot{\varphi} = \Omega_0^2 \left[ \left( 1 + \frac{a_V}{g} \right) \sin \varphi + \frac{a_H}{g} \cos \varphi - \frac{1}{mgL} \left( k_L \varphi - k_N \varphi^3 \right) \right], \quad \Omega_0 = \sqrt{\frac{gL}{i^2 + L^2}}. \quad (4)$$

For moderately large rotations, this non-linear ODE may be simplified by truncating the Taylor expansions for trigonometric terms as follows:

$$\sin \varphi = \varphi - \frac{1}{6} \varphi^3 + ..., \quad \cos \varphi = 1 - \frac{1}{2} \varphi^2 + .... \quad (5)$$

The corresponding cubic non-linear ODE reads

$$\ddot{\varphi} = \Omega_0^2 \left[ \frac{a_H}{g} + \left( \frac{a_V}{g} + 1 - \frac{k_L}{mgL} \right) \varphi - \frac{1}{2} \frac{a_H}{g} \varphi^2 + \left( \frac{k_H}{mgL} - \frac{1}{6} \left( \frac{a_V}{g} + 1 \right) \right) \varphi^3 \right]. \quad (6)$$

This equation will be treated as benchmark problem subsequently.
Its linearized version has the form

$$\ddot{\varphi} = \Omega_\varphi^2 \left[ -\frac{1}{\kappa} \frac{a_H}{g} + \left( 1 - \frac{1}{\kappa} \frac{a_V}{g} \right) \varphi \right], \quad (7)$$

where the squared linear frequency of the pendulum with a linear restoring spring is given by

$$\Omega_\varphi^2 = \Omega_0^2 \kappa, \quad \kappa = \frac{k_L}{mgL} - 1. \quad (8)$$

The primary instability regime for vertical parametric resonance, see, e.g., ([17], Chapter 23) and ([23], Chapter 10), extends from the following frequency

$$\Omega_V = 2\Omega_\varphi = 2\Omega_0 \sqrt{\kappa}. \quad (9)$$

For the harmonic excitation at this parametric resonance frequency, we obtain the following non-linear second-order ODE, assuming the horizontal ground excitation frequency also to be in resonance, i.e., to be equal to the vertical parametric resonance frequency:

$$\ddot{\varphi} = \Omega_0^2 \left[ \bar{a}_H(t) \left( 1 - \frac{1}{2} \varphi^2 \right) + \left( \bar{a}_V(t) + 1 \right) \left( \varphi - \frac{1}{6} \varphi^3 \right) - \frac{k_L}{mgL} \varphi + \frac{k_N}{mgL} \varphi^3 \right] = f_2(\varphi, t),$$

$$\bar{a}_H(t) = \frac{a_{H0} \sin(\Omega_V t)}{g}, \quad (10)$$

$$\bar{a}_V(t) = \frac{a_{V0} \sin(\Omega_V t)}{g}.$$

Amplitudes of horizontal and vertical earthquake accelerations are denoted as $a_{H0}$ and $a_{V0}$, respectively. It is convenient to convert Equation (10) into the normal form

$$\begin{pmatrix} \dot{\varphi} \\ \dot{\omega} \end{pmatrix} = \begin{pmatrix} f_1(\omega) \\ f_2(\varphi, t) \end{pmatrix},$$

$$f_1(\omega) = \omega. \quad (11)$$

We now present a discussion of the computational method published in [11] and how to apply it in the present context. Our method is a time-stepping procedure; in the following, we consider time intervals of equal length $T$. The normal form of the balance equation in Equation (11) is brought into its integral form by integrating over the finite interval (time step) $\tau \in [0, T]$, the local time in the time interval being defined as $\tau = t - (n-1)T$, where $n = 1, 2, \ldots$ is the number of the time step under consideration, and $t$ denotes the

physical time, starting at the beginning of the observation period. We thus obtain the following formal representation of the angular velocity for any local time $\tau$ in the interval

$$\omega(\tau) = \int_0^\tau \tilde{f}_2(\varphi(\bar{\tau}), \bar{\tau}) d\bar{\tau} + \omega_0. \tag{12}$$

The reason for assigning $f_2$ formally with a superimposed tilde is explained later, after Equation (15). The angle of the tower from the vertical direction then follows as

$$\varphi(\tau) = \int_0^\tau f_1(\omega(\bar{\tau})) d\bar{\tau} + \varphi_0. \tag{13}$$

The generally inhomogeneous initial conditions at the beginning of the time interval under consideration are:

$$\varphi(0) = \varphi_0, \quad \dot{\varphi}(0) = \omega_0. \tag{14}$$

The two integral Equations (12) and (13) are solved in an iterative manner, motivated by the work of Picard [13]. We however perform the first iterative step by substituting into the integral in Equation (12) the result obtained for $\varphi$ from suitable explicit discrete-mechanics-type scheme. As a reference one, we here take the explicit fourth-order Runge–Kutta time-integration scheme, see [16]:

$$\begin{aligned}
&\varphi(\bar{\tau}) = \varphi_0 + (k_{11} + 2k_{21} + 2k_{31} + k_{41})\bar{\tau}/6, \\
&k_{11} = f_1(\omega_0), \ k_{21} = f_1(\omega_0 + k_{12}\bar{\tau}/2), \ k_{31} = f_1(\omega_0 + k_{22}\bar{\tau}/2), \ k_{41} = f_1(\omega_0 + k_{32}\bar{\tau}), \\
&k_{12} = f_2(\varphi_0, 0), \ k_{22} = f_2(\varphi_0 + k_{11}\bar{\tau}/2, \bar{\tau}/2), \ k_{32} = f_2(\varphi_0 + k_{21}\bar{\tau}/2, \bar{\tau}/2), \\
&k_{42} = f_2(\varphi_0 + k_{31}\bar{\tau}, \bar{\tau}).
\end{aligned} \tag{15}$$

An analogous result is obtained for $\omega(\bar{\tau})$, which is not repeated for the sake of brevity. Having substituted Equation (15) into Equation (12), the integration in Equation (12) is not performed directly, but is carried out via a truncated Taylor series representation of the Runge–Kutta approximation for $\varphi(\bar{\tau})$ of Equation (15). The number of Taylor series terms may change from iteration step to iteration step. Thus, the tilde for $\tilde{f}_2$ in Equation (12) refers to truncated Taylor series of some prescribed order. The integral in Equation (13) is solved by substituting $\omega$ of Equation (12), where $\tau$ is replaced by $\bar{\tau}$. The necessary analytic operations, such as Taylor-series representations and integration, can be easily performed by means of symbolic computation. Since we approximate the relations of balance in their integrated (global balance) form, and not directly in their differential one and since we start the iteration using the well-established explicit fourth-order Runge–Kutta scheme, convergence is generally fast; for large free non-linear vibrations of the pendulum, see the detailed comparative study in [11]. The necessary analytic operations of the procedure, such as Taylor-series representations and integration, can be easily performed by means of symbolic computation. At the end of iteration, we obtain an analytic formula for the time evolution of $\varphi(\tau)$ and $\omega(\tau)$ during the time interval $\tau \in [0, T]$. Setting $\tau = T$ yields the values at the end of the time interval, $\varphi(T)$ and $\omega(T)$. In a time stepping procedure, these values serve as initial values for the next time step. The analytic results obtained in the first time step after the end of iteration can be directly utilized, where $\tau$ now means local time in the next time step, starting again at the beginning of the latter.

A main advantage of this technique is that the Picard-type iteration and the analytic operations necessary for performing it must be done only once, for the first time interval, i.e., before the time-stepping procedure starts. Analytic expressions for $\varphi$ and $\omega$ then have been obtained by means of symbolic computation, which can be stored and used in every time step. The considered numbers of Taylor terms, the number of iterations, and the length $T$ of the time steps serve as parameters of the numerical problem, which allows adjusting the cost/accuracy ratio of the proposed computation for a specific problem at hand.

### 3. Oscillations under Harmonic Earthquake Excitation in Both Directions

For a realistic computation, we take structural parameters that we estimated from a special tower-like structure discussed in Chapter 23 of the monograph by Petersen and Werkle [17]: $i = 8.66$ m is the radius of inertia, $L = 15$ m is the distance of mass, $mg = 3600$ kN is the total gravity force, $k_L = 3 \times 10^{10}$ Nm is the linear stiffness coefficient, and we take $k_N = 0.9 \times 10^{10}$ Nm a non-linear stiffness coefficient. Moreover, we use $a_{H0} = 9.81$ m/s$^2$ as the horizontal acceleration amplitude, $a_{V0} = 2.94$ m/s$^2$ as the vertical acceleration amplitude, and $T = 0.1$ s is the considered time step length.

We perform the computations using the method described above (Picard-type iterations in Mathematica) and compare with the numerical standard function NDSolve of Mathematica, see [12]. The algorithm is described in Appendix A.1. The comparison is given in Figure 2. The results of both methods agree well. The proposed scheme has an advantage in the computation time: 0.20 s versus 0.94 s. In the derivation of the explicit formula used to plot Figure 2, in our method we implement three subsequent Picard-type iterations with 4, 6, and 9 Taylor terms, respectively, and compile the derived formula using the procedure Compile of Mathematica. Increasing the number of Picard iterations and the terms in the Taylor expansion would not change the relative error further in a visible manner. We refer the reader interested in technical details how the results are affected by the number of terms of Taylor expansion and by the time step length to Appendix A.2.

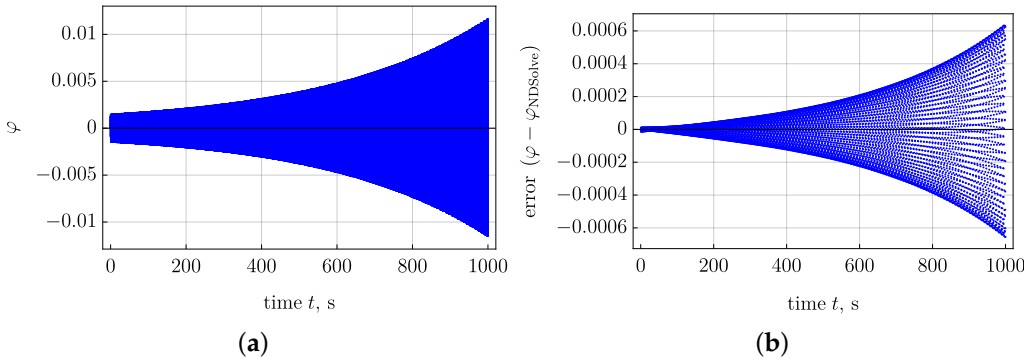

**Figure 2.** (**a**) Angle $\varphi$ over physical time $t$ for harmonic earthquake excitation in both directions $a_{H0} = 9.81$ m/s$^2$: our method, the computation time is 0.20 s (three iterations with (4, 6, 9) Taylor terms (for NDSolve of Mathematica the time history is visually indistinguishable and the computation time is 0.94 s). (**b**) Absolute deviation between solutions with our method and NDSolve.

### 4. Numerical Results for Real Earthquakes

To perform the numerical computations, we again use our method and the standard initial value problem solver NDSolve of Mathematica, see [12]. We exemplarily use a ground acceleration record from the Kobe earthquake, considered to be one of the most devastating and costly natural disasters in recent history [24]. The Kobe earthquake data, originally given as equally spaced acceleration values, on which our computations are based, are shown in Figure 3. For the NDSolve computations, earthquake data are interpolated by means of the standard procedure Interpolation of Mathematica (the interpolation order is set to 0, which means a piecewise-constant function at the output), see also Figure 3. The values from the earthquake data are used as constants in the corresponding time steps. For simplicity sake, the vertical excitation is taken proportional to the horizontal one, $a_V = 0.3a_H$. The set of Kobe earthquake data has the time step $T = 0.01$ s. The number of time steps used in the computations is 4090. We use two iterations with 7 Taylor terms each and with standard machine precision in the computations and compile the resulting symbolic formulas using the procedure Compile of Mathematica.

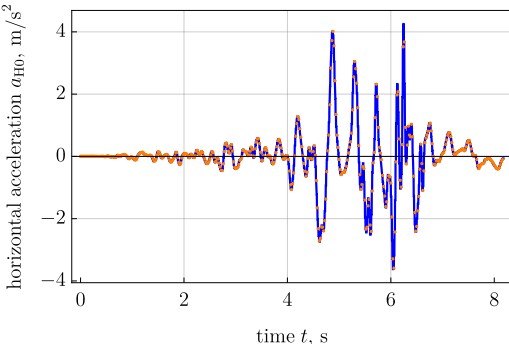

**Figure 3.** Horizontal acceleration over physical time *t* for Kobe earthquake data interpolated with Mathematica (an initial data segment with largest accelerations is shown). The orange points denote the Kobe earthquake data and the blue curve is for the interpolated function.

The results of the numerical integration in the case of real earthquake record in Figure 3 are plotted in Figure 4, which shows the comparison between the results of our method and the NDSolve procedure of Mathematica. We observe well agreement between the results, where the computation time is substantially shorter for the proposed method. An analogous behavior was found when using records for other disastrous earthquakes, such as the El Centro earthquake and the Bam earthquake. We note, however, that our choice of interpolating earthquake data as piecewise-constant, which was motivated by the practical requirement not to add further functional assumptions to the point-wise measured data, may impose significant performance penalty to NDSolve. Namely, it forces this algorithm to use extremely short steps at the jumps. In contrast, for the present Picard-type method the latter jumps do not result in problems, because, as mentioned already above, differentiability of the integrands is not required, only existence of the integrals.

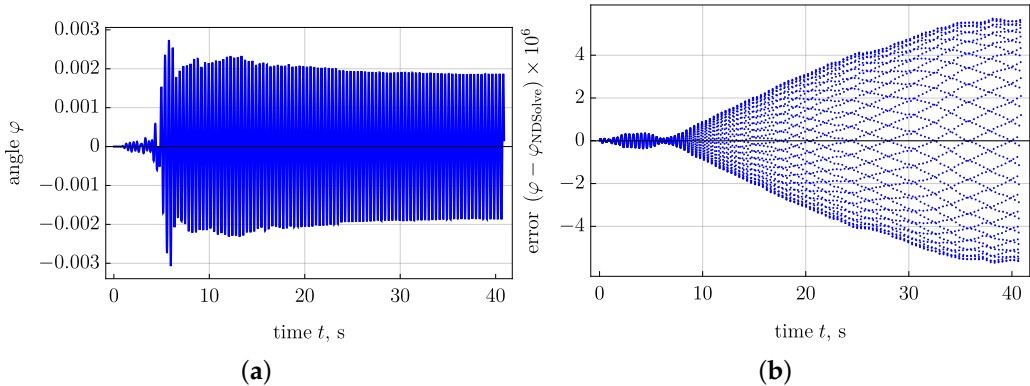

**Figure 4.** (**a**) Angle $\varphi$ over time *t* for Kobe earthquake data with proposed method, the computation time is 0.40 s (for NDSolve of Mathematica the time history is visually indistinguishable and the computation time is 1.45 s). (**b**) Absolute deviation between solutions with our method and NDSolve.

## 5. Double Pendulum

In this section, we investigate free vibrations of the double pendulum using the above presented method and compare the results with the analytic solutions. The scheme of the double pendulum is given in Figure 5.

The system of equations reads

$$\ddot{\varphi}_1 + \alpha \varphi_1 - \beta \varphi_2 = 0,$$
$$\ddot{\varphi} - \gamma \varphi_1 + \gamma \varphi_2 = 0.$$
$$\alpha = \Omega^2 \left(1 + \frac{m_2}{m_1}\right), \quad \beta = \Omega^2 \frac{m_2}{m_1}, \quad \gamma = \alpha \frac{L_1}{L_2},$$

(16)

where the dimensionless time is introduced as

$$T = \Omega t, \quad \Omega \equiv \sqrt{\frac{g}{L_1}}. \tag{17}$$

The problem is linear and admits the exact solution in the form

$$\varphi_1(t) = \frac{\beta}{\delta}\varphi_2(0)\left[-\cosh\left(\sqrt{\frac{-\alpha-\gamma-\delta}{2}}t\right) + \cosh\left(\sqrt{\frac{-\alpha-\gamma+\delta}{2}}t\right)\right],$$

$$\varphi_2(t) = \frac{1}{2\delta}\varphi_2(0)\left[\begin{array}{l}(-\alpha+\gamma+\delta)\cosh\left(\sqrt{\frac{-\alpha-\gamma-\delta}{2}}t\right) + \\ +(\alpha-\gamma+\delta)\cosh\left(\sqrt{\frac{-\alpha-\gamma+\delta}{2}}t\right)\end{array}\right], \tag{18}$$

$$\delta = \sqrt{\alpha^2 - 2\alpha\gamma + 4\beta\gamma + \gamma^2}$$

in the special case with the following initial conditions

$$\varphi_1(0) = 0, \quad \dot{\varphi}_1(0) = \dot{\varphi}_2(0) = 0. \tag{19}$$

For the double pendulum we first apply a discrete-mechanics-type scheme by Greenspan [11,15], subsequently denoted as Greenspan I, as the first guess in the Picard-type iteration. The calculation technique consists of the following steps:

$$\begin{aligned}F_{10} &= -(\alpha - \beta)\varphi_{10} - \beta(\varphi_{10} - \varphi_{20}),\\F_{20} &= \gamma(\varphi_{10} - \varphi_{20}),\\\omega_i &= \omega_{i0} + \tau F_{i0},\\\varphi_i &= \varphi_{i0} + \frac{\omega_i + \omega_{i0}}{2}\tau.\end{aligned} \tag{20}$$

The last two formulas are then integrated iteratively four times with the integral form of the balance of momentum and with the integral kinematic relation

$$\omega_i = \omega_{i0} + \int_0^\tau F_i d\bar{\tau},$$

$$\varphi_i = \varphi_{i0} + \int_0^\tau \omega_i d\bar{\tau}, \tag{21}$$

where

$$\begin{aligned}F_1 &= -(\alpha - \beta)\varphi_1 - \beta(\varphi_1 - \varphi_2),\\F_2 &= \gamma(\varphi_1 - \varphi_2).\end{aligned} \tag{22}$$

In this section, we use the computer algebra system Maple [22] for symbolic computations and plotting. In Figure 6, we draw the relative single-step error with the Greenspan I scheme for the double pendulum. The terms of the approximate solution with $k$ iterations are the same up to the $2k + 2$ terms of the Taylor expansion of the exact solution. Even for large time steps is the error rather small.

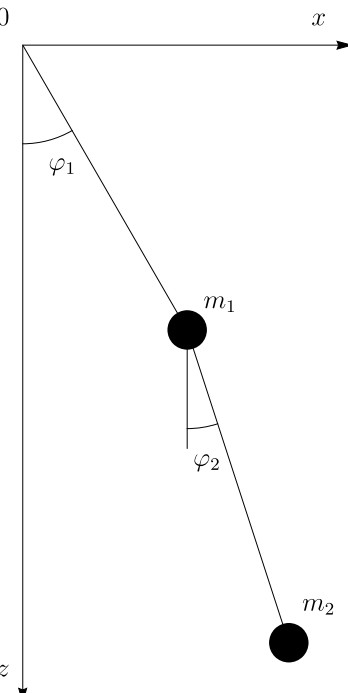

**Figure 5.** Scheme of double pendulum.

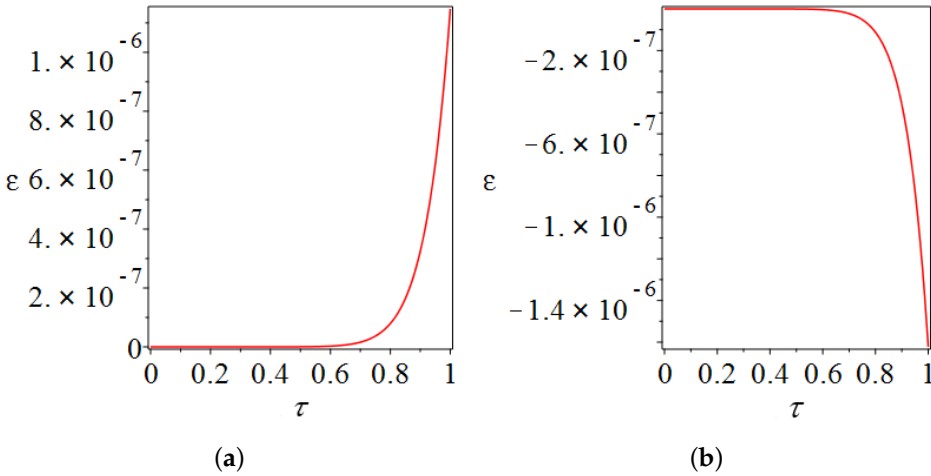

(a)                                                          (b)

**Figure 6.** Error $\varepsilon = (\varphi_i - \varphi_i^{\text{analytic}})/\varphi_{20}$ of Greenspan I method for (**a**) $\varphi_1$ and (**b**) $\varphi_2$ angles for a single time step. $\tau = 1$ means $\approx 5$ largest periods and $\approx 12$ smallest periods.

We also apply the second discrete-mechanics-type Greenspan scheme (Greenspan II) [11,15] given by

$$
\begin{aligned}
F_{10} &= -(\alpha - \beta)\varphi_{10} - \beta(\varphi_{10} - \varphi_{20}), \\
F_{20} &= \gamma(\varphi_{10} - \varphi_{20}), \\
\omega_{i1} &= \omega_{i0} + \frac{F_{i0}}{2}\tau, \\
\varphi_{i1} &= \varphi_{i0} + \frac{\omega_{i1} + \omega_{i0}}{4}\tau, \\
F_{11} &= -(\alpha - \beta)\varphi_{11} - \beta(\varphi_{11} - \varphi_{21}), \\
F_{21} &= \gamma(\varphi_{11} - \varphi_{21}), \\
\omega_{i} &= \omega_{i1} + \frac{\frac{3}{2}F_{i1} - \frac{1}{2}F_{i0}}{2}\tau
\end{aligned}
\tag{23}
$$

with the mid-point formula for the angle

$$\varphi_i = \varphi_{i0} + \frac{\omega_{i0} + \omega_i}{2}\tau. \tag{24}$$

The result is then iteratively integrated four times with the balance relation and with the kinematic relation.

In Figure 7 we present the relative errors in the angles for the Greenspan II scheme. We see in the plots that the relative errors are again small even for larger time steps. The terms of the approximate solution with $k$ iterations are the same up to the $2k + 2$ terms of the Taylor expansion of the exact solution. There are further terms in the approximate solution, which doe not exactly correspond to the exact solution. The errors of four iterations are presented; the results appear to be converging to the exact solution as expected.

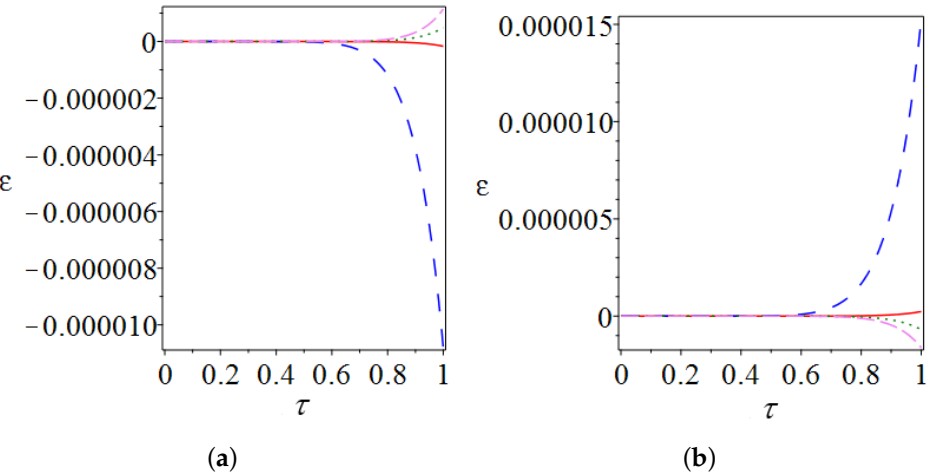

**Figure 7.** Error $\varepsilon = (\varphi_i - \varphi_i^{\text{analytic}})/\varphi_{20}$ of Greenspan II method for (**a**) $\varphi_1$ and (**b**) $\varphi_2$ angles for a single time step. $\tau = 1$ means $\approx 5$ largest periods and $\approx 12$ smallest periods.

Our last task is to present the relative errors in a multistep procedure composed from the Greenspan II scheme with four iterations. The results are given in Figure 8 for a large number of oscillations corresponding to $n =$ 10,000 time steps.

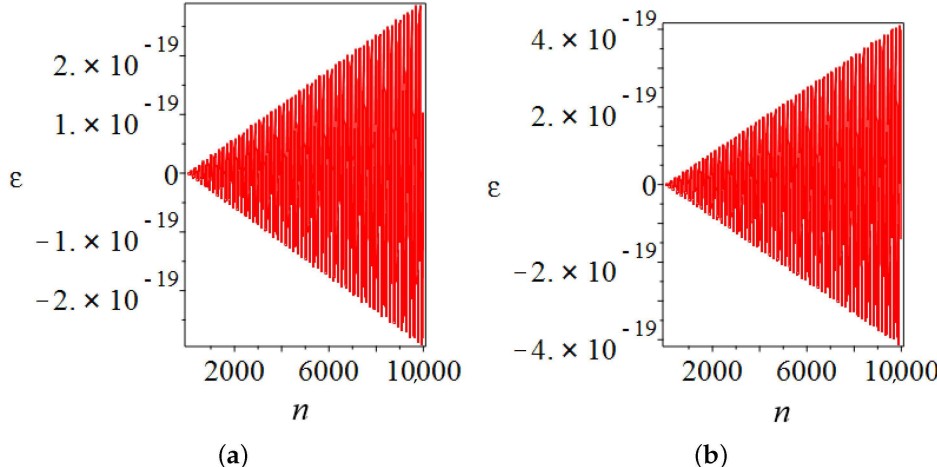

**Figure 8.** Error $\varepsilon = (\varphi_i - \varphi_i^{\text{analytic}})/\varphi_{20}$ of Greenspan II method over timestep index for (**a**) $\varphi_1$ and (**b**) $\varphi_2$ angles obtained in a multistep procedure with the fixed time step $T = 0.02$ (the dimensionless periods of the natural oscillations are 0.0861 and 0.208).

## 6. Conclusions

In the above exemplary study, we showed the application of our discrete-mechanics-type time-integration technique with extended Picard-type iterations, first published in [11], to the example of an earthquake-excited tower-like structure, modeled as a rigid non-linear pendulum. The corresponding vibration problems were also solved using the standard ND-Solve method Wolfram Mathematica [12], and the outcomes were compared successfully to our method, where our proposed approach required considerable less computation time. Free vibrations of the double pendulum were studied afterwards in order to demonstrate that the method can be applied with advantage under linear conditions also, leading to a high accuracy even for comparatively large time-steps. Since our explicit time-integration method is based on analytic expressions, which can be obtained by symbolic computation in a straight-forward manner, it is hoped that it will find interest in various scientific engineering applications, e.g., in earthquake engineering, in vibration control, and for hybrid testing.

**Author Contributions:** Conceptualization, E.O. and H.I.; methodology, E.O. and H.I.; software, E.O.; validation, E.O.; formal analysis, E.O. and H.I.; investigation, E.O. and H.I.; writing—original draft preparation, E.O. and H.I.; writing—review and editing, E.O. and H.I.; visualization, E.O.; supervision, H.I.; project administration, H.I.; funding acquisition, H.I. All authors have read and agreed to the published version of the manuscript.

**Funding:** This research has been supported by the COMET-K2 Center of the Linz Center of Mechatronics (LCM) funded by the Austrian federal government and the federal state of Upper Austria. Open Access Funding by the University of Linz.

**Institutional Review Board Statement:** Not applicable.

**Informed Consent Statement:** Not applicable.

**Data Availability Statement:** Not applicable.

**Acknowledgments:** Authors are grateful to the reviewers for their highly constructive remarks, e.g., concerning additional citations and various technical aspects, such as in connection with NDSolve.

**Conflicts of Interest:** The authors declare no conflict of interest.

## Appendix A. Technical Details for Calculation with Harmonic Excitation

*Appendix A.1. Algorithm*

1. Prescribe initial data.
2. Define $\Omega_V$ using Equation (9).
3. Prescribe general initial conditions of the IVP.
4. Prescribe the time step length and the number of time steps.
5. Define $f_1$ and $f_2$ usingEquation (11).
6. Define the Runge–Kutta coefficients and formulas using Equation (15).
7. Define the function that symbolically integrate polynomials.
8. Symbolically integrate the truncated version of the Runge–Kutta formula.
9. Add initial velocity $\omega_0$ to obtain the first iteration of $\omega$.
10. Symbolically integrate $\omega$ and add initial angle $\varphi_0$ to obtain the first iteration of $\varphi$.
11. Repeat integration to obtain a more accurate formula; in the present case, we use 3 iterations with (4, 6, 9) Taylor terms.
12. Truncate the obtained formula by the third order of the Taylor series in initial conditions $\omega_0$ and $\varphi_0$ of each time step.
13. Compile the explicit formulas of time integration.
14. Perform the iterative time-stepping method.
15. Prepare graphs.

*Appendix A.2. Effects of the Number of Terms of Taylor Expansion and of the Time Step*

**Table A1.** Effect of the number of terms of Taylor expansion on the deviation and computation time.

| Numbers of Taylor Terms | Computation Time | Maximum Deviation |
|---|---|---|
| 3, 5, 8 | 0.20 | $7.3 \times 10^{-3}$ |
| 4, 6, 9 | 0.20 | $6.3 \times 10^{-4}$ |
| 5, 7, 10 | 0.26 | $6.9 \times 10^{-4}$ |
| 6, 8, 11 | 0.28 | $6.8 \times 10^{-4}$ |
| 7, 9, 12 | 0.30 | $6.7 \times 10^{-4}$ |

**Table A2.** Effect of the time step on the deviation and computation time.

| Time Step Length | Number of Time Steps | Computation Time | Max. Deviation |
|---|---|---|---|
| 0.2 | 5000 | 0.096 | 0.013 |
| 0.1 | 10,000 | 0.20 | $6.3 \times 10^{-4}$ |
| 0.05 | 20,000 | 0.45 | $2.4 \times 10^{-5}$ |
| 0.025 | 40,000 | 0.80 | $2.0 \times 10^{-5}$ |

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
