# Peer review of "Application of a Novel Picard-Type Time-Integration Technique to the Linear and Non-Linear Dynamics of Mechanical Structures: An Exemplary Study"

_applsci, doi:10.3390/app11093742_

Round 1

Reviewer 1 Report

The paper is well presented and discussed. It can be accepted in its present form

Author Response

Authors are grateful to the reviewer for his/her support of our paper.

Reviewer 2 Report

I have read the paper carefully and found it enough novel and interesting. The paper was written well in English and no serious error has been found. From a mathematical point of view, the mathematical relations are clearly presented and the related results are arranged well. The only shortcoming of the article, from this reviewer’s point of view, is the low number of literature on the topic of linear as well as nonlinear dynamic analysis. In this case and for dynamic analysis of advanced systems and structures, the literature review can be expanded using the following papers (not limited to these);

https://doi.org/10.1016/j.soildyn.2014.07.013,

https://doi.org/10.1155/2013/237370,

https://doi.org/10.1080/15397734.2016.1277740

https://doi.org/10.1002/mma.6758,

https://doi.org/10.1061/(ASCE)0733-9445(1991)117:7(2035), https://doi.org/10.1016/j.compstruc.2012.03.019,

https://doi.org/10.3390/sym12040643,

https://doi.org/10.1016/j.ijengsci.2020.103371,

DOI: 10.1109/AIM.2019.8868689

Author Response

The authors would like to express their sincere thanks to the Reviewers for their interest, the encouraging words and for the constructive remarks. We have tried to consider these remarks as far as this appeared to be possible, given the necessarily tight publication schedule of the SI. Consideration of these remarks however has lead to a considerable improvement of the revised version, for which we are very grateful to the Reviewers. We use red colors to indicate changes in the following.

Reviewer 2: “The only shortcoming of the article, from this reviewer’s point of view, is the low number of literature on the topic of linear as well as nonlinear dynamic analysis....” The following sentences and citations have been included in the revised version:

Introduction, after “…see, e.g., [4], for a force correction method.”:

For the development of dynamic structural models in linear and non-linear earthquake engineering, as well as for suitable time-integration methods and non-linear control formulations, the reader is moreover referred exemplarily to [5new] - [10new].

[5new] S. Nagarajaiah, A. M. Reinhorn and M. C. Constantinou, Nonlinear Dynamic Analysis of 3DBaseIsolated Structures. ASCE Journal of Structural Engineering, Vol. 117, 2035-2054 (1991)

[6new] Kurt Schlacher, Andreas Kugi, Hans Irschik, Nonlinear control of earthquake excited high raised buildings by approximate disturbance decoupling. Acta Mechanica, Vol. 125, 49-62 (1997)

[7new] Hans Irschik, Kurt Schlacher, Andreas Kugi, Control of earthquake excited nonlinear structures using Liapunov's theory. Computers and Structures, Vol.67, 83-90 (1998)

[8new] T. Liu, Ch. Zhao, Q. Li and L. Zhang, An efficient backward Euler time-integration method for nonlinear dynamic analysis of structures. Computers & Structures, Vol: 106, 20-28 (2012)

[9new] B. Atmaca, M. Yurdakul, S. Ateş, Nonlinear dynamic analysis of base isolated cable-stayed bridge under earthquake excitations. Soil Dynamics and Earthquake Engineering, Vol. 66, Page: 314-318 (2014)

[10new} S. Dastjerdi, B. Akgöz, Ö. Civalek, M. Malikan, and V. A. Eremeyev, On the non-linear dynamics of torus-shaped and cylindrical shell structures. Journal of Engineering Science, Vol. 156, Page: 103371 (2020)

Introduction, after “…For small vibrations of the earthquake-excited pendulum with an inelastic spring considering the P-Delta effect, see [12].”:

For some other advanced applications of pendulum vibrations in the present context, see, e.g., [19new] and [20new].

[19new] Najeeb Alam Khan, Nadeem Alam Khan, F. Riaz, Dynamic Analysis of Rotating Pendulum by Hamiltonian Approach. Chinese Journal of Mathematics, vol. 2013, Article ID 237370, 4 pages (2013)

[20new] F. Mazza, S. Sisinno, Nonlinear dynamic behavior of base-isolated buildings with the friction pendulum system subjected to near-fault earthquakes. Mechanics Based Design of Structures and Machines, Vol. 45, 331-344 (2017)

Reviewer 3 Report

The authors present a time-integration technique to the non-linear and linear dynamical problems, which is based on the Picard iteration method. After a general introduction, a mechanical model of a tower-like structure is derived in Sect 2, the method is illustrated using two examples (an inverse pendulum, a double pendulum) comparing numerically obtained results with a theoretical or a reference numerical solution.

Unfortunately, the method itself is not described in detail, which is a bit challenging for the reader. A pseudocode/algorithm listing would be greatly welcomed.

It also seems that the method is not suitable for general equations. Could you comment on its applicability? E.g., in a case when the explicit integration is difficult/impossible?

The topic is adequate for the journal, the paper is carefully prepared and well written; yet it needs some significant improvements. The plots are clear and readable, however, their informational merit can be improved.

The paper can be considered for publication after major changes.

General comments:

\begin{itemize}

\item p4, Eq. (10). The common excitation frequency $\Omega_V$ is used for both directions?

\item p4, Eq. (12). What is $\tilde{f}_2$? (I see, definition is on the next page.)

\item p4, line -3. \textit{The two integral Eqs. (12) and (13) are solved in an iterative manner ...} Could you please indicate iterations, e.g., by indices? Is there any hint for a number of iterations? (e.g., stopping criterion)

\item p5, Eq. (15). It seems that the relation for $\omega(\tilde{\tau})$ is missing. (Although from further text it seems not to be used). Mind also $k_1$ at the first line.

\item p5, line 124-125 \textit{The results of both methods agree well ... } The actual agreement is not visible from Figure 2. Plot the relative error instead of the solution time history. How does it change for different number of Picard iterations, terms of the Taylor expansion, time step?

\item p5, line 125 \textit{...irregularities in the right plots are because of the large number of plotted points...} Use \texttt{PlotPoints} or \texttt{PerformanceGoal} options for the \texttt{Plot} command.

\item p6, line 137 \textit{For the NDSolve computations, earthquake data are interpolated... piecewise-constant ...} Actually, such a choice imposes significant performance penalty to the \texttt{NDSolve}, which uses mostly an adaptive time stepping and discontinuous right hand side forces the algorithm to use extremely short steps at the jumps.

\item p6, line 140-141 \textit{For the numerical solution with our technique in Mathematica, the values from the earthquake data are used as constants in the corresponding time steps.} (Missing verb in the sentence.) It seems that the sampling rate was 500Hz, i.e., $\Delta t=0.002$s and thus for one step with $T=0.1$ there were 50 samples. How were computed the integrals Eq. (12), (13)?

\end{itemize}

Author Response

The authors would like to express their sincere thanks to the Reviewers for their interest, the encouraging words and for the constructive remarks. We have tried to consider these remarks as far as this appeared to be possible, given the necessarily tight publication schedule of the SI. Consideration of these remarks however has lead to a considerable improvement of the revised version, for which we are very grateful to the Reviewers. We use red colors to indicate changes in the following.

Reviewer 3:

Unfortunately, the method itself is not described in detail, which is a bit challenging for the reader. A pseudocode/algorithm listing would be greatly welcomed.”

We have added the algorithm listing in Appendix A.1.

It also seems that the method is not suitable for general equations. Could you comment on its applicability? E.g., in a case when the explicit integration is difficult/impossible?” The following sentences have been included in the revised version:

Introduction, after: “Our technique consists in an application of an extended Picard iteration to the time-integrated (global balance) form of the equations of motion, see [7] for the original Picard iteration, and [8] for the structural equations of motion in integral form, i.e., for the global relations of balance of momentum.”:

Concerning the applicability of this technique, we believe that it will be suitable as long as the derivation of the equations of motion has lead to a system of linear or non-linear ordinary differential equations of second order for the required degrees-of-freedom (DOFs), accompanied by the necessary number of initial conditions. In structural mechanics, such a initial-value system can be obtained by starting from the relations of (linear and/or angular) momentum for suitably modelled substructures (rigid bodies, deformable finite elements after discretization), taking into account constitutive relations, and using the usual reduction techniques to obtain a system with minimum number of DOFs. The system afterwards must be formally time-integrated to a system of first order. The time-integrated system then represents a system of so-called the global relations of balance of momentum, or first integrals. Often, when possible, it is more advisable to use the latter from the scratch, due to their wider applicability, since only the existence of integrals are to be required from a mathematical point of view, and not differentiability, or bounded integrands. Only mild mathematical assumptions hence must be made for allowing to substitute approximations into the integrands of these global balance forms of the equations of motion to obtain an improved solution, as this was originally suggested by Picard [8]. In the following, in extension of Picard [8], explicit discrete mechanics-type approximations, see Greenspan [9].....

p4, Eq. (10). The common excitation frequency is used for both directions? The following explanation has been included in the revised version:

For the harmonic excitation at this parametric resonance frequency, we obtain the following non-linear second-order ODE, assuming the horizontal ground excitation frequency also to be in resonance, i.e. to be equal to the vertical parametric resonance frequency:

p4, Eq. (12). What is ~ f2? (I see, definition is on the next page.).” The following clarification has been included in the revised version after Eq. (12):

The reason for assigning f2 formally with a superimposed ~ is explained later, after Eq. (15).

p4, line -3. The two integral Eqs. (12) and (13) are solved in an iterative manner ... Could you please indicate iterations, e.g., by indices? Is there any hint for a number of iterations? (e.g.,stopping criterion). The following sentence has been included on p5 in the revised version after “…convergence is generally fast; for large free non-linear vibrations of the pendulum, see the detailed comparative study in [5].:

In the following non-linear examples of forced earthquake-type excitations, we needed only up to three iterations and up to 9 Taylor terms, in order to obtain results that agree well with the comparative numerical solutions.

p5, Eq. (15). It seems that the relation for omega is missing. (Although from further text it seems not to be used). Mind also k1 at the first line.”

The parameter k11 has been added. The following explanation has been included after Eq. (15):

An analogous result is obtained for omega(tau..), which is not repeated for the sake of brevity.

p5, line 124–125 The results of both methods agree well ... The actual agreement is not visible from Figure 2. Plot the relative error instead of the solution time history. How does it change for different number of Picard iterations, terms of the Taylor expansion, time step?”

See the deviation between results in Figs. 2 and 3 and the effects of the number of terms of Taylor expansion and of the time step length on the deviation and computation time in Appendix A.

The following explanation has been included after Fig. 2: Increasing the number of Picard iterations and terms in the Taylor expansion would not change the relative error further in a visible manner. We refer the reader interested in technical details how the results are affected by the number of terms of Taylor expansion and by the time step length to Appendix A.2.

p5, line 125 ...irregularities in the right plots are because of the large number of plotted points...”

Figure has been improved according to the suggestion of the Reviewer, and the sentence has been removed, thank you!

p6, line 137 For the NDSolve computations, earthquake data are interpolated... piecewise-constant.

... Actually, such a choice imposes significant performance penalty to the NDSolve, which uses mostly an adaptive time stepping and discontinuous right hand side forces the algorithm to use extremely short steps at the jumps.” The following sentence has been added after “as the El Centro earthquake and the Bam earthquake.”:

We note however, that our choice of interpolating earthquake data as piecewise-constant, which was motivated by the practical requirement not to add further functional assumptions to the point-wise measured data, may impose significant performance penalty to NDSolve. Namely, it forces this algorithm to use extremely short steps at the jumps. In contrast, for the present Picard-type method the latter jumps do not result in problems, because, as mentioned already above, differentiability of the integrands is not required, only existence of the integrals.

p6, line 140-141 For the numerical solution with our technique in Mathematica, the values from the earthquake data are used as constants in the corresponding time steps. (Missing verb in the sentence.) The sentence has been simplified to:

The values from the earthquake data are used as constants in the corresponding time steps.

It seems that the sampling rate was 500Hz, i.e., t = 0.002s and thus for one step

with T = 0.1 there were 50 samples. How were computed the integrals Eq. (12), (13)?

We should clarify the misunderstanding: our method uses the time step length T = 0.1 for the results plotted in Figure 2, and a large time step sufficient for computations is one of its main advantages; no subdivisions of this interval are required. The integrals in Eq. (12) and (13) are computed symbolically as described in Appendix A.1.

We have also added the following sentence to Acknowledgments: Authors are grateful to the Reviewers for their highly constructive remarks, e.g. concerning additional citations and various technical aspects, such as in connection with NDSolve.

Round 2

Reviewer 3 Report

I would like to thank the authors for the new improved version. It looks out well now. I wish I had more free time to study the topic in a greater detail.

Let me add just two more comments which are perhaps not essential for publication. I hope it can help you in your further work.

1. The error plot  in Figs 2 and 4 is much better than just visual comparison used before, however, it can be improved further. E.g.,  plotting an (relative!) error of maximal values of the amplitudes and phase shifts of the both solutions.

2. A common flaw in engineering approach to a numerical solution to ODEs, namely when discrete measured data are used, is neglect of the fact  that the (higher order) numerical schemes impose an assumption on continuity of the input data. This may be correct or not depending on the sampling rate and character of the data.  In your case, however, you mix an assumption on the 4th order continuity in the rk4 step (initial approximation), which is complemented by an explicit integration which uses only piecewise constant function. I wonder if this subtlety may have any influence to the total accuracy.